# Hierarchical lifelong topic modeling using rules extracted from network communities

**Muhammad Taimoor Khan**[1]*, **Nouman Azam**[1], **Shehzad Khalid**[2], **Furqan Aziz**[3]

**1** Department of Computer Science, National University of Computer and Emerging Sciences, Peshawar Campus, Peshawar, Pakistan, **2** Department of Computer Engineering, Bahria University, Islamabad, Pakistan, **3** Centre for Computational Biology, University of Birmingham, Birmingham, United Kingdom

* taimoor.khan@nu.edu.pk

**Data Availability Statement:** The data underlying the results presented in the study are available from (Computer Science Department, University of Illinois Chicago at https://www.cs.uic.edu/~zchen/downloads/KDD2014-Chen-Dataset.zip).

## Abstract

Topic models extract latent concepts from texts in the form of topics. Lifelong topic models extend topic models by learning topics continuously based on accumulated knowledge from the past which is updated continuously as new information becomes available. Hierarchical topic modeling extends topic modeling by extracting topics and organizing them into a hierarchical structure. In this study, we combine the two and introduce hierarchical lifelong topic models. Hierarchical lifelong topic models not only allow to examine the topics at different levels of granularity but also allows to continuously adjust the granularity of the topics as more information becomes available. A fundamental issue in hierarchical lifelong topic modeling is the extraction of rules that are used to preserve the hierarchical structural information among the rules and will continuously update based on new information. To address this issue, we introduce a network communities based rule mining approach for hierarchical lifelong topic models (NHLTM). The proposed approach extracts hierarchical structural information among the rules by representing textual documents as graphs and analyzing the underlying communities in the graph. Experimental results indicate improvement of the hierarchical topic structures in terms of topic coherence that increases from general to specific topics.

## Introduction

Topic models process a collection of documents to extract hidden thematic structures called topics [1, 2]. Each topic is an ordered lists of contextually correlated words representing a concept. Topic modeling has many extensions based on the learning paradigm such as supervised, semi-supervised, transfer learning, hybrid and knowledge-based models. These extensions are designed to meet the specific demands and needs found in different application areas [3–5]. In this paper, we examine a new type of topic model called hierarchical lifelong topic models.

Lifelong topic models (LTM) is an important extension to topic models where topics are continuously learned and refined based on previously accumulated knowledge and processing of new information [3, 6]. Unlike other extensions, it does not require external guidance to analyze a dataset or to mine rules from it. More specifically, it exploits the huge volume of data

**Funding:** Please be informed that we haven't received any funding during this research work. However, National University of Computer and Emerging Sciences will award faculty honorarium when the paper is published. There was no additional external funding received for this study.

**Competing interests:** The authors have declared that no competing interests exist.

processed in the past and the overlap in their concepts to identify global patterns [7]. Such an approach is highly desired in an unsupervised analysis of continuously arriving data from diverse sources [8]. Hierarchical topic modeling is another exciting extension which organizes the topics in a hierarchical structure thereby allowing the topics to be viewed at different levels of granularity. The topics at higher levels are abstract and get more specific down the hierarchy [9]. The incorporation of hierarchical topics in lifelong topic models will allow for examining the topics at different levels of granularity and adjusting the granularity as new information becomes available. This may be useful in applications such as summarization of news stories at different levels based on continuous streaming of news items [10].

A fundamental issue for incorporating hierarchical topics in lifelong topic models is the efficient and effective extraction of rules that are used to construct topics. Existing approaches are not suitable for this task. Although, the rule extraction approaches in LTM models facilitate the production of compact and coherent topics. They are, however, not well suited to arrange the topics in a hierarchy [8, 11–13]. In particular, they provide little or no information regarding the structural relationships that may occur among the topics [14, 15]. On the other hand, the rule extraction approaches in hierarchical topic models do not benefit from the past experience and rely on some level of supervision or manual guidance. In addition to this, whenever new information is available, the same support is required. To overcome these issues, we present an approach called network communities based rule mining for hierarchical lifelong topic models or NHLTM.

The proposed NHLTM approach extract rules that provides the hierarchical structure among the rules and update the hierarchical relationships among the rules whenever more information becomes available. In particular, rules are extracted by constructing a graph based on the data in textual documents and then communities of related words are detected within the graph using spectral clustering. Experimental results on datasets of Chen 2014 electronics having 50 categories, Chen 2014 non-electronic having 50 categories and Reuters 21,578 having 90 categories suggest topic hierarchies with improved topics interpretability [16, 17]. The hierarchical structure of the topics is rationalized by gradual improvement of topic coherence across the topics at different levels. The proposed approach shows an improvement in topic coherence as we move down the hierarchy. The empirical log-likelihood of the held-out test indicates stable behavior of the model for predicting the unseen data at varying levels and number of topics in the hierarchy.

## Topic models, lifelong and hierarchical topic models

### Topic models

A topic consist of contextually related words that collectively represents a concept [18, 19]. An inference technique is typically used that iteratively determines the distribution of words across different topics based on words relationships in documents [20]. Words that co-occur across many documents have high chance of being selected under the same topic [21]. Topic models are used in different applications to explore, categorize, summarize and analyze textual data [22, 23].

Topic models have been extended in variety of ways and lead to models such as supervised models, hybrid models, transfer learning models, semi-supervised models, knowledge-based models and lifelong learning models [19, 24, 25]. In supervised models, the documents in training data are tagged with manually provided set of topic labels [26, 27]. Semi-supervised topic models incorporate expert guidance in associating certain words to topics while the other words are grouped around them in their respective topics [7]. Hybrid topic models are trained on certain features using a small labeled data and is provided for unsupervised analysis

on remaining data. It is used in case of limited availability of labeled training data. The supervised and hybrid topic models help to produce interpretable topics, however, its use is only limited to the datasets for which such labeled training data exists. Transfer learning based topic models are trained on one dataset and applied on another having limited or noisy documents [28, 29]. Knowledge-based topic models are manually provided with set of rules that constraint the model to observe them while extracting topics [11, 30, 31]. The inference technique i.e., Gibbs sampling is modified in knowledge-based topic models where it has to incorporate the rules impact as well, as external guidance, in deciding the associations among words for a topic. Lifelong learning approach with topic modeling can be considered as the knowledge-based topic modeling where the rule mining mechanism is automated [32].

Hierarchical topic modeling uses some level of supervision or external guidance to combine the topics of a dataset into hierarchical structure [33]. The high level topics that are closer to root are general topics while the topics away from root are specialized topics. The literature of lifelong topic modeling and hierarchical topic modeling is presented below and their shortcomings are highlighted.

## Lifelong topic modeling

Lifelong topic models continue to mine new rules from the dataset of each task, while it benefits from the relevant rules that were mined in the previous tasks [32, 34]. Considering the diversity of the datasets processed in the past, only the rules that are relevant to the current task can be of help [3, 6, 15, 35]. For this purpose, the background or context of a rule is compared with the vocabulary of the current dataset. It may be considered as resolving polysemy at the rule level. Although the quality of automatically mined rules may not be as good as that of a human expert, however, it is compensated by the quantity of rules and the freedom that it can be applied to data of any nature [12, 36, 37].

Generally speaking, the lifelong learning cycle contains four main components [38]. The first component deals with mining rules from a task completed. The rules are first evaluated based on some evaluation criteria reflecting their quality. Important rules are mined based on suitable thresholds on the evaluation criteria. The rules may represent semantic or contextual correlations among words [31, 39]. The approaches used for evaluating the quality of rules in literature are Normalized point-wise mutual information, frequent itemset mining, multi-support frequent itemset mining etc [8, 12, 13]. This components executes after performing each task of extracting topics from a dataset.

The second component deals with representing the rules in a format where they can be efficiently and effectively retrieved when required [31]. In literature, the rules are represented as word pairs with positive correlation and strength of correlation, called must-link rules. For example, the must-link rule $mustLink(word_1, word_2, +, 0.67)$ indicates a positive correlation between the words $word_1$ and $word_2$, indicated by the + sign. This rule adds bias into the gibbs sampling based inference to increase the probability of $word_2$ for topics where $word_1$ has higher probability and vice versa with an impact of 0.67 [32]. Similarly, the rules with word pairs having negative correlation with a strength of negative correlation is called cannot-link rules. For example, the cannot-link rule $cannotLink(word_1, word_3, -, 0.86)$ indicates a negative correlation between the words $word_1$ and $word_3$, shown by the − sign. The rule adds a bias into the gibbs sampling based inference to decrease the probability of $word_3$ for topics where $word_1$ has high probability and vice versa with an impact of 0.86 [8, 12, 34]. Thus, in the modified gibbs sampling the intuition of arranging words into topics comes from the current dataset and relevant rules of the past [34].

The third component deals with the retention and maintenance of rules to preserve them for future [32]. The rules are maintained with different levels of abstraction, providing a trade-off between storage space and efficiency [8, 12, 34, 36]. Lifelong learning can also be considered as a longer chain of transfer learning with unknown target datasets, multiple source datasets and that can be processed in any sequence. Therefore, it demands perseverance of rules for longer duration. For example, the rules learnt from $2^{nd}$ task may be utilized by the $10^{th}$ or $50^{th}$ task. On the other hand, the rules mined automatically are expected to have wrong, irrelevant, duplicate and even contradictory rules. Therefore, the retention and maintenance module undergo a refining process to resolve these issues and keep only consistent and good quality rules.

The fourth component deals with the transfer of rules relevant to the current task and benefiting from it by incorporating the bias from the rules [11]. The gibbs sampling inference technique is extended with the Generalized polya urn (GPU) model [12]. The GPU model and its other variants are responsible for introducing automatic external guidance for different types of rules into the inference of topic models [30, 32, 34, 36, 40]. In recent times, word embeddings have improved various natural language processing approaches and is successfully used with topic models [41, 42]. Large text corpora have issues of long tailed vocabularies that result in losing the context. Using topic models on word embeddings helps better grouping of words into topics as topic models also attempt to group words based on their context [43]. Topic embeddings are also used to represent topics as dense vectors and help evaluate the correlation among topics [41]. Embeddings are not introduced in the current LTM approaches and may prove to be very effective in preserving context of words within topics and topics within a dataset.

## Hierarchical topic modeling

Hierarchies are crucial in displaying the various components of a system in a tree-like structure. It has generic concepts at higher levels, while specific concepts are at lower levels [33]. Topics that are structured in a hierarchy can be analyzed at different levels of granularity and compared to one another [44]. There are some studies on hierarchical topic modeling with traditional topic modeling approaches [9, 28, 33]. We refer the approaches that are not using lifelong learning approach as the traditional approaches. They perform hierarchical topic modeling for a dataset in isolation.

The hLDA model is the earliest attempt in organizing topics into a hierarchical structure [14, 45]. It make use of the nested chinese restaurant process (nCRP) to set a prior on possible hierarchical trees using the topic prior $\beta$. The gibbs sampling technique is also modified to predict new topic for the sampled word in the given document while it also predict the level of topics associated with the same document. The SSHLDA model organize the topics of a dataset into a hierarchical structure using labeled training data [9]. Transfer learning based topic models are used to transfer the hierarchical structure of one dataset into another related dataset [28]. It may be noted that hierarchical document topic modeling is also a closely related area where the topic hierarchies are extracted by representing the textual documents in the form of a graph. There are many useful approaches based on hierarchical document topic modeling including [46–48]. The essential assumption in these approaches is that the explicit linkage among documents in order to arrange them into a hierarchical structure [49, 50]. Although this assumption is useful in many cases, however, in our study, we do not make any such assumptions. We only rely on words and their co-occurrence in order to represent the given textual data in the form of a graph. The words co-occurrence is generally vague due to large heavy-tailed vocabularies but useful patterns can be identified when combined over a number of datasets.

The notable attempts for extracting topic hierarchies in literature either require external support or hyper-parameters for the hierarchical structure which limits their application to certain scenarios only. Moreover, they perform hierarchical topic modeling in isolation and would require the same support each time. On the other hand, lifelong topic models benefit from the topic modeling performed on the previous datasets to improve the coherence of topics for the relevant future datasets. But the existing lifelong topic modeling approaches make use of statistical measures to mine rules. These approaches mine, maintain and transfer the rules in isolation, which makes them inconclusive towards generating topic hierarchies [8, 12, 13]. Therefore the existing lifelong topic modeling approaches are not well suited for finding the associations among topics [51, 52]. In particular, they lack in conveying structural information which is a key constituent for extracting topic hierarchies [52, 53]. In the next section, we demonstrate this limitation of the existing rule mining approaches and indicate towards a possible solution.

## A limitation in existing rule mining approaches for topic modeling

In this section, we highlight a limitation in rule mining approaches that are used in existing lifelong topic modeling approaches. Consider Table 1 which represents a sample dataset. The rows of the table correspond to documents and the columns correspond to words. A "-" in a cell means that the corresponding word appears in the respective document. For instance, $RAM$ is present in document $D_1$, $D_2$, $D_5$, $D_6$, $D_7$, $D_8$ and $D_9$. We are interested in co-occurrences of words across the documents. Table 2 is constructed for this purpose based on data in Table 1. The rows and columns of Table 2 correspond to the words. A particular entry in the table reflects the co-occurrence between a pair of words. For instance, a value of 6 in the second column corresponding to the first row means that the words $RAM$ and $Cache$ has co-occurred in 6 documents. This can be easily seen in Table 1, i.e., $RAM$ and $Cache$ both occurred together in Documents $D_1$, $D_2$, $D_5$, $D_6$, $D_8$ and $D_9$.

The typical lifelong topic modeling approaches consider rules in the form of word pairs. Among the rules, only those are selected whose evaluation is at or above a certain threshold value. Lets assume that the words co-occurrence is used as an evaluation measure with a threshold being set to 5. This will result in selection of four rules, namely, $r_1(RAM, Cache)$, $r_2(Intel, Core)$, $r_3(Price, Cost)$ and $r_4(Graphics, Video)$. Since these rules have a co-occurrence frequency of 5 or more. We may note that the rule are evaluated in isolation, irrespective of their associations with other rules. More specifically, it ignores the relationships among

**Table 1. Sample dataset with 10 documents containing 9 words.**

|          | RAM | Cache | Intel | Core | Price | Grapahics | Video | keyboard | Cost |
|----------|-----|-------|-------|------|-------|-----------|-------|----------|------|
| $D_1$    | -   | -     | -     | -    |       |           |       |          |      |
| $D_2$    | -   | -     | -     | -    |       | -         | -     | -        |      |
| $D_3$    |     |       | -     | -    | -     |           |       |          | -    |
| $D_4$    |     | -     | -     | -    | -     |           |       | -        | -    |
| $D_5$    | -   | -     | -     | -    |       | -         | -     |          |      |
| $D_6$    | -   | -     |       |      |       | -         | -     |          |      |
| $D_7$    | -   |       | -     | -    | -     | -         | -     | -        | -    |
| $D_8$    | -   | -     |       |      | -     |           |       |          | -    |
| $D_9$    | -   | -     |       |      | -     |           |       | -        | -    |
| $D_{10}$ |     |       |       |      |       | -         | -     | -        |      |

A dash (-) indicates the presence of a word in a document.

**Table 2. Term-term matrix based on the data in Table 1.**

|  | RAM | Cache | Intel | Core | Price | Graphics | Video | Keyboard | Cost |
|---|---|---|---|---|---|---|---|---|---|
| RAM | - | 6 | 4 | 4 | 3 | 4 | 4 | 3 | 3 |
| Cache |  | - | 4 | 4 | 2 | 3 | 3 | 3 | 3 |
| Intel |  |  | - | 6 | 3 | 3 | 3 | 3 | 3 |
| Core |  |  |  | - | 3 | 3 | 3 | 3 | 3 |
| Price |  |  |  |  | - | 1 | 1 | 2 | 5 |
| Graphics |  |  |  |  |  | - | 5 | 3 | 1 |
| Video |  |  |  |  |  |  | - | 3 | 1 |
| Keyboard |  |  |  |  |  |  |  | - | 3 |
| Cost |  |  |  |  |  |  |  |  | - |

different rules. For example, the rules $r_1$ and $r_2$ have words having very strong association such as *RAM* and *Intel* having a co-occurrence value of 4. This relationship is however not being considered in the existing approaches.

The issue of mining rules in isolation can be intuitively addressed by considering the association among the rules in the form of co-occurrence of the words belonging to different rules. We define the associations based on co-occurrence information between words belonging to different rules as,

$$Association(r_i, r_j) = \frac{\sum_{w \in r_i, w' \in r_j} CF(w, w')}{|r_i| \times |r_j|}. \tag{1}$$

where $w$ and $w'$ are distinct words belonging to different rules $r_i$ and $r_j$, respectively. The association between two rules is the summation of all word co-occurrences across the two rules. For instance, consider the rules $r_1$ and $r_2$, the association between these two rules according to Eq (1) is,

$$Association(r_1, r_2) = \frac{CF(RAM, Intel) + CF(RAM, Core) + CF(Cache, Intel) + CF(Cache, Core)}{|RAM, Cache| \times |Intel, Core|}$$
$$= \frac{4 + 4 + 4 + 4}{2 \times 2} = 4 \tag{2}$$

The association among rules are calculated for all the other rules as well and are given in Table 3. These associations can be used to find which rules are more closely related to each other as compared to others. Considering all the rules as leaf nodes of a tree, the rules $r_1$ and $r_2$ have the highest association of 4 according to Table 3. They are linked at the next higher level to form a more general rule. We call this the rule $r_{1,2}$, which can also be considered as the parent rule for the two rules $r_1$ and $r_2$. Now at the next higher level, there are three rules i.e., $r_{1,2}$, $r_3$ and $r_4$. Following the same, the association among these rules is evaluated where the rules $r_{1,2}$ and $r_4$ have the next highest association with an association score of 3.25. The association

**Table 3. Values of association among the rules based on Eq 1.**

|  | $r_1$ | $r_2$ | $r_3$ | $r_4$ |
|---|---|---|---|---|
| $r_1$ |  | 4 | 3 | 3.5 |
| $r_2$ |  |  | 3 | 3 |
| $r_3$ |  |  |  | 1 |
| $r_4$ |  |  |  |  |

score of rules $r_{1,2}$ and $r_3$ is 3 and that of rules $r_3$ and $r_4$ is only 1. Thus, the rule $r_{1,2,4}$ is the next level parent rule that has the rules $r_{1,2}$ and $r_4$ as its descendants. Finally, the two available rules i.e., $r_{1,2,4}$ and $r_3$ having association score of 2.3 are linked to form a single rule of $r_{1,2,3,4}$. This is one way of finding the structural association among rules.

Exploiting the structural association among rules, a hierarchical structure may be generated for the topics of a given dataset. Different approaches are available for finding the association among rules that could be translated into improving the arrangement of topics within a dataset and that of words within a topic. For instance, consider topics $T_1, T_2, \ldots T_K$. The topic $T_1$ is containing the words of the rule $r_1$ and $T_2$ is containing the words of the rule $r_2$. Since the rules $r_1$ and $r_2$ have the closest association, therefore, topics $T_1$ and $T_2$ are merged at the higher level to represent a single topic $T_{1,2}$. The structure between the topics can be successively generated by combining the topics containing rules with highest associations. This structure will lead to hierarchy with the dataset as root node and the specific topics as leaf nodes while the intermediate nodes representing general topics at different levels. The model has incomplete knowledge, and therefore some topics may not be represented by rules. Such topics link to the root of the hierarchy as outliers. The number of such topics depends on the number of relevant rules in the knowledge base. They are expected to decrease as the number of rules and their diversity increases with experience. It may be noted that the existing studies generally consider rules in the form of word pairs only. We consider rules in the form of word pairs and in addition to that we also consider rules that may contain any number of words.

A critical issue in the above intuitive solution is how to determine and model the association of rules based on sound theoretical notions. The exploration of this issue will open up new ways for exploiting the relationships among the rules. Graph based community detection is widely used for exploratory data analysis in many areas [54]. Representing the data as graph and performing the graph Laplacian enhances the clustering properties in the data. Such properties are generally not found in conventional clustering algorithms [55, 56]. For any type of data represented as a graph, community detection techniques can help in detecting the communities and their hierarchical structures from the data [57, 58]. These approaches have been used successfully for research problems in the fields of psychology, sociology, biology, statistics and computer science [57, 59, 60]. Community detection has been used to identify clusters of similar articles using a citation network [61]. In this paper, we explore the use of community detection technique to mine structurally associated rules.

## Network communities based hierarchical lifelong topic modeling (NHLTM) approach

In this section, we present our proposed NHLTM approach. The lifelong learning aspect allows the model to preserve rules of diverse nature over experience. The proposed community detection approach for mining rules facilitate in improving coherence of topics and arrange them into a hierarchical structure.

### Architecture of NHLTM approach

The Architecture that realizes the NHLTM approach is presented in Fig 1. A dataset is provided to this architecture as input. Hierarchically structured topics are produced from the given dataset as output. The rules mined from the datasets processed in the past contribute towards organizing words into topics and topics into a hierarchical structure. Our contribution in this research is to introduce a community detection based approach named NHLTM to mine structurally associated rules. The architecture initiates the rule mining module whenever a new dataset is made available. The other modules i.e., rule representation and rule transfer

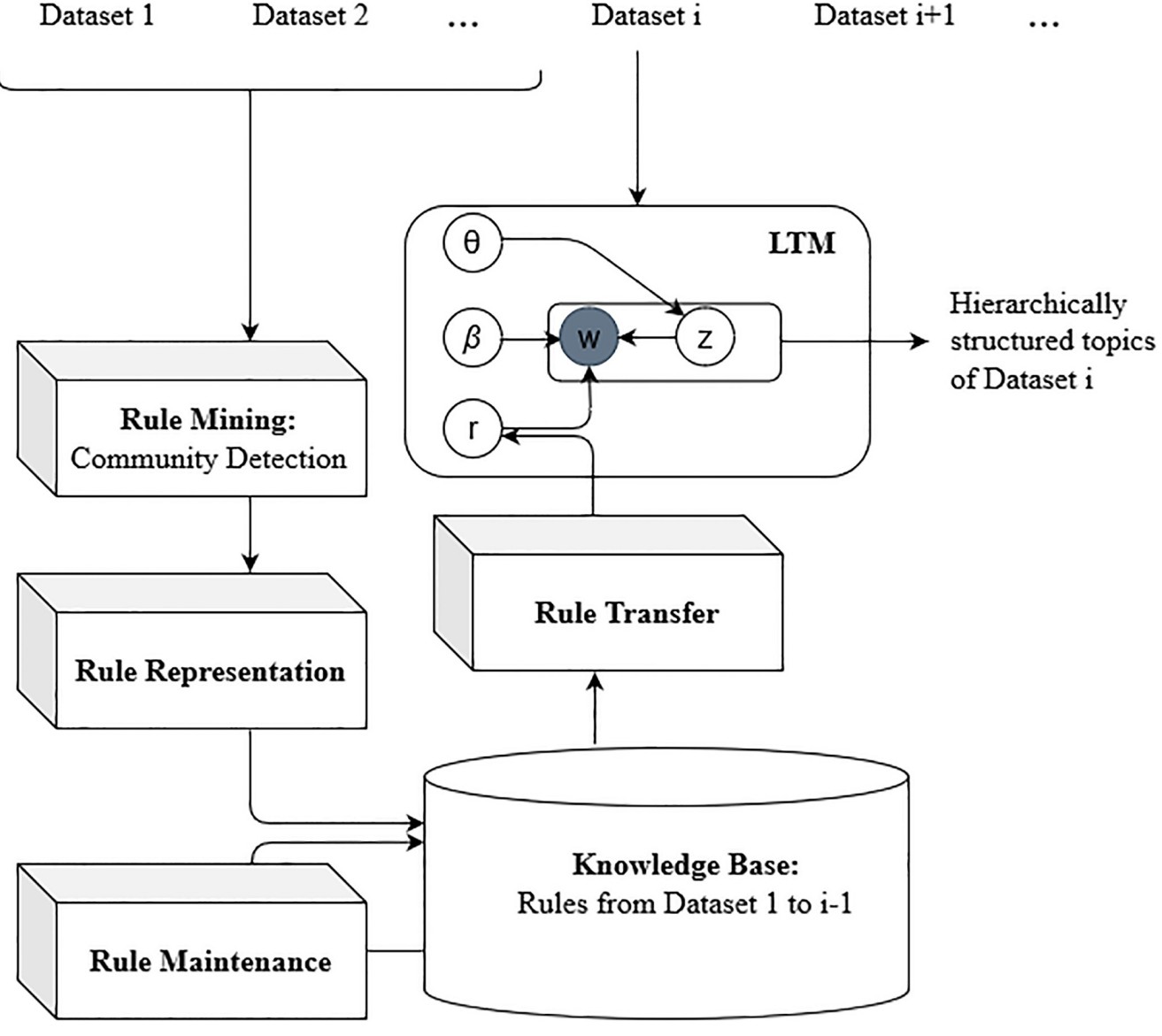

**Fig 1. Architecture of NHLTM approach.**

are also updated based on new datasets which helps in efficient and effective utilization of the newly introduced community based rules. The knowledge base retains all the rules for long term future use while the maintenance module ensures the refinement of rules based on new data.

## Community detection based rule mining

In order to perform community detection, the following four step process is generally used [56, 62]. The first step is to represent the given data as a graph. The next step is to recursively split communities into smaller and smaller communities until the stopping criteria is met. Generally, the communities at leaf nodes have strong correlation and represent contextually correlated words. There may be few exceptions with leaf node communities representing noise

words in the dataset. The third step is to evaluate communities and discard those that do not have enough compactness among words. The last step is to only retain those communities that have an acceptable level of compactness as correlation among its words. Our proposed NHLTM approach uses the aforementioned steps of community detection for mining rules.

The details of NHLTM approach is given in Algorithm 1. It takes the current dataset $D^i$ and a graph $G$ corresponding to all previously processed datasets $D^1$ to $D^{i-1}$ as an input. The output of the algorithm are the updated topic hierarchies and the updated graph $G$. The algorithm first extract the relevant knowledge rules from the graph $G$ of previously processed datasets. The while loop in the algorithm corresponds to the mechanism of transferring the impact of rules into LTM which relates to the final step of the NHLTM approach. In line 10, the structural similarity among rules are evaluated based on similarity in the assigned hexadecimal codes to organize them into a hierarchical structure. Lines 11—14 are related to the lifelong learning components of the NHLTM approach. We now elaborate the community detection process for rule mining employed in the algorithm in some detail.

**Algorithm 1** Proposed NHLTM Approach

```
Input: Dataset D^i, G of previously processed datasets from D^1 to D^{i-1}
Output: Topic Hierarchy for D^i, updated Graph G
1: Get relevant rules from prior knowledge rules
2: LTM random initialization
3: while Words are not settled in their respective topics (Inference)
do
4:    Predict the most appropriate topic t for the sampled word w
according to the current state
5:    Update the word-topic and topic-document distributions
accordingly
6:    if w in rules then
7:       Increase the probability of all words w′ in topic t that are in
rule r ∈ rules with the word w
8:    end if
9: end while
10: Extract Topic Hierarchy of Dataset D^i, using the structural asso-
ciation codes of rules
11: Add words as nodes and their co-occurrence as edges from D^i to
Graph G of D^1 to D^{i-1}
12: Extract communities using spectral clustering
13: Filter communities with weak intra-community associations
14: Represented the communities as rules
15: Generate new rules i.e., rules_{D_i} and add to the knowledge base
16: return Topic Hierarchies, Graph G
```

**Step 1**: **Representing the data as graph**

The algorithm 1 constructs a graph $G$ from the textual data by considering words as nodes and their co-occurrence as edges. In particular, the textual data corresponding to datasets $D^1$ to $D^{i-1}$ are used for this purpose. It is an undirected graph as the edges represent the words co-occurrence that can be interpreted both ways. The edges are weighted highlighting the co-occurrence frequency of the two words forming an edge. Considering the power law distribution in the graph where few words have very high occurrence frequency as compared to others and therefore, are more frequent in stable co-occurrence associations. Such graphs can be categorized as scale-free networks that require the weaker filtering the weaker associations. Network based representation has many useful applications in text analytics including keywords extraction, document categorization and knowledge graphs [63]. In this paper, the network of words across all datasets represent a hierarchical knowledge graph that assists topic modeling improving topic coherence and generating hierarchical structure of topics within a dataset.

The weaker nodes and edges are filtered using minimum node degree and minimum edge weight thresholds respectively. The advantage of dropping weak edges having low weights and weak nodes having low degrees is to reduce the computational cost. Since the graph is expected to grow in size, only reliable rules are extracted. The intuition for clipping weak branches is that they are not going to make strong rules due to their weak associations. They usually represent noise or rare vocabulary that is not frequently used by other users [56, 64].

**Step 2**: **Identifying and detecting word communities**

Communities can be extracted using different community detection algorithms [54, 55]. For extracting communities, we use the Fiedler eigenvector. It represents the nodes with positive and negative values, indicating a possible split as the nodes of positive values may form one sub-community while the others may form another sub-community. It is obtained by representing the graph $G$ as a normalized Laplacian matrix and decomposing it into eigenvectors [65]. The conductance cut is next used to evaluate if it is feasible to proceed with splitting the graph into communities and sub-communities recursively. The evaluation is based on the cost associated with inter-community associations.

It is worth mentioning that there are many community detection algorithms that can be used for extracting communities however, we used spectral clustering for this purpose. The edges within a community help in identifying useful and important rules while the edges between communities help in identifying the hierarchical association among the rules. This approach continuously updates the communities based on new information which results in updated hierarchical information among the rules. Spectral clustering performs better for dense graphs and has reasonable results in various text processing tasks. The graph of multiple datasets with pruning weak nodes and their edges ensures that the graph is dense, thus making spectral clustering more favorable. Spectral clustering algorithms are based on computing eigenvalues that allow considering different levels of information and have dimensionality reduction feature which is a typical issue in textual processing tasks. All this makes spectral clustering a more favorable approach for our study.

**Step 3**: **Evaluating compactness of communities**

The communities extracted in step 2 are next evaluated based on the associations among the nodes contained in the community. The measure of weighted connectivity index or *WGCI* is used for this purpose [55]. It measures the intra-community associations in relation to the inter-community relations. Nodes with higher intra-community associations have higher *WGCI* score as the intra-community edge weights increases proportional to the degree of nodes. In contrast, the nodes with high inter-community associations have low Weighted graph connectivity index *WGCI* as the intra-community edges will have low weights and their respective nodes have high degrees. The *WGCI* of a community $comm_i$ can be calculated as,

$$WGCI(comm_i) = \sum_{e_{w,w'}} \frac{weight(e_{w,w'})}{\sqrt{deg(w) \times deg(w')}} \tag{3}$$

Where $e_{w,w'}$ is the edge between the words $w$ and $w'$ belonging to the community $comm_i$. In addition to *WGCI*, we also consider the measure of Von Neumann entropy for evaluating communities [66] as,

$$Entropy(S) = -\sum_{n=1}^{|N|} \frac{\hat{\lambda}_n}{2} ln \frac{\hat{\lambda}_n}{2} \tag{4}$$

where N is the total number of nodes in the community and $\hat{\lambda}_n$ represents their eigenvalues of the normalized adjacency matrix corresponding to the second lowest value. By putting suitable

thresholds on the two measures, we selected communities that have considerable associations among its nodes. The threshold values are experimentally chosen to keep only those communities that have higher intra-community associations [67].

**Step 4**: **Representing communities as rules**

The communities selected are abstracted and represented as rules. The objective is to facilitate efficient storage, retrieval and transfer of rules into a task [68]. The words in a community are retained as the rule words which also helps in setting up the context for the rules. The structural information of a community is retained in the rule by preserving the order of splits that lead to the generation of each community. The impact of a rule is calculated using Lins Approximation [55]. The value of Lins Approximation is aggregated across all edges of the community to give a measure of entropy for the community.

**Step 5**: **Transferring the impact of rules into Gibbs sampling**

This step deals with transferring the impact of relevant rules into the Gibbs sampling inference of topic distribution for the current task. The default inference technique continuously update the document-topic and topic-word distributions. It starts from a random state and converges on the suitable distribution in an iterative way [18]. The distributions in a particular iteration refer to the state of the model, where new state is determined based on the previous state following markov chain [69]. The probability of a suitable topic for the sample word is [70],

$$P(z_i = t|z^{-i}, w, d, \alpha, \beta) \propto \theta_{d,t} \times \phi_{t,w} \tag{5}$$

where a sampled word $w$ from document $d$ has its probability updated for topic $t$ as $\theta_{d,t}$ times $\phi_{t,w}$. The $\theta_{d,t}$ and $\phi_{t,w}$ are given by,

$$\theta_{d,t} = \frac{\eta_{d,t}^{-i} + \alpha}{\sum_{k=1}^{T}(\eta_{d,k}^{-i} + \alpha)} \tag{6}$$

and

$$\phi_{t,w} = \frac{\eta_{t,w}^{-i} + \beta}{\sum_{u=1}^{V}(\eta_{t,u}^{-i} + \beta)} \tag{7}$$

respectively. In the Eq (6), the document-topic distribution i.e., $\theta_{d,t}$ is updated using $\eta_{d,t}^{-i}$ as the probability of topic $t$ in document $d$ based on the current state of the model. It is computed by excluding the sampled occurrence of the word, indicated by $-i$. $\alpha$ is the smoothing factor. The denominator normalizes the value using the sum of probabilities of all the other topics from $k = 1, \ldots, T$ for the same document $d$. Similarly in Eq (7), the topic-word distribution i.e., $\phi$ is updated using $\eta_{t,w}^{-i}$ which is the probability of word $w$ in topic $t$ based on the current state of the model. It is again computed by excluding the sampled occurrence of the word $w$, indicated by $-i$. Moreover, $\beta$ is used as a smoothing factor. The denominator normalizes the value using the sum of probabilities of all the words from $u = 1, \ldots, V$ for the same topic $t$.

In order to select relevant rules for a task, the overlap of a rule is measured with the vocabulary of the task and are compared against a threshold. It is given by [12] as,

$$\frac{\#(V \cap r)}{\#r} > \xi_{overlap} \tag{8}$$

where $\xi_{overlap}$ is the threshold for overlap between rule words and the vocabulary of the current task. It helps to resolve the context and address the word sense disambiguation for a rule.

The bias from the rules effects the topic-word distribution i.e., $\phi_{t,w}$ only. When the probability of a word $w$ is increased for a topic $t$ and $w$ is part of a rule $w \in r$, then the other words of the same rule $w' \in r$ also have their probabilities increased for the same topic $t$, using equation given by [12] as,

$$\phi_{t,w',r} = \frac{v_{r_{w,w' \in r}} \times \eta_{t,w'}^{-i} + \beta}{\sum_{u=1}^{V} (v_{r_{w,w' \in r}} \times \eta_{t,u=w'}^{-i} + \beta)} \qquad (9)$$

where the probabilities of all other words $w' \in r$ are updated for topic $t$ by the $v_r$ factor of the rule when $w \in r$ has its probability updated for $t$. However, in case $w \notin r$, the default Eq (7) will be used to update $\phi_{t,w}$.

## Experimental results and discussion

In this section, we present detailed experimental results of the proposed NHLTM model for a sequence of tasks. The Chen 2014 dataset with 50 electronic product categories, the Chen 2014 dataset with 50 non-electronic product categories and Reuters R21578 with 90 categories are considered for this purpose [16, 17]. These datasets have documents in multiple categories assumed to be available in a sequence and are used for lifelong topic modeling in literature. In case of the first two datasets, i.e., Chen 2014 electronic and Chen 2014 non-electronic, there are more than 5000 in each category. Each document is a short product review that is collected from Amazon.com. The third dataset has skewed document representation with categories having documents in thousands, hundreds, and below that. Each category is considered a dataset for extracting topic hierarchies and is processed in a sequence. Therefore, it fits more into the context of mining rules continuously as an automatic process. Lifelong topic modeling cannot be applied to a single dataset as it requires multiple tasks to improve with experience.

Table 4 shows the details of the graphs for each of the three datasets. Only 10% of the past data is considered for mining rules to reduce computational cost. The datasets are preprocessed to retain only meaningful words as verbs, adverbs, adjectives, and nouns. The words as nodes, their edges and corresponding weights are mentioned before and after graph filtering. Through aggressive filtering, the weak nodes and edges are removed as they are least likely to be part of compact communities. It results in increasing the average node degree and edge weight. The filtering mechanism refines the graph for efficient analysis. It drops the nodes and edges that have a low degree and edge weight, respectively. Thus we have a smaller but more densely connected network after the filtering phase.

In the second step, the graph is processed to identify and extract possible communities. The Chen 2014 electronic dataset has a total 148 communities extracted that are interrelated to

**Table 4. Graph properties before and after pruning weak nodes and edges from the dataset.**

| Dataset | Property | Before Filtering | After Filtering |
|---|---|---|---|
| Electronic Chen 2014 | Word Nodes | 5,574 | 1,228 |
| | Co-occurrence Edges | 6,55,261 | 3,46,687 |
| | Per Edge Weight | 3.43 | 4.84 |
| Non-Electronic Chen 2014 | Word nodes | 7,391 | 2,572 |
| | Co-occurrence Edges | 5,83,020 | 4,24,066 |
| | Per Edge Weight | 2.19 | 2.49 |
| Reuters R21578 | Word Nodes | 5,193 | 2470 |
| | Co-occurrence Edges | 9,74,793 | 6,73,193 |
| | Per Edge Weight | 1.62 | 1.82 |

**Table 5. Sample communities with their codes, top words and neighboring communities.**

| ID (Hexa Codes) | Top Words | Neighboring Community ID |
|---|---|---|
| 9 | Adaptor, cam, filter, stand, indicator | C |
| C | Exception, weird, superior, break | 9 |
| 56 | Field, section, flat | 8A |
| 8A | Storage, airport, convenient | 56 |
| 21E | Environment, rest, world, answer | 23D |
| 23D | Folder, label, credit | 21E |
| 439 | Profile, background, scene, rock | 43B |
| 43B | Tray, blueray, manual, gripe | 439 |
| 10A9 | Alarm, longer, beep | 43B |
| 2146 | Police, ticket, state, ka | 2151 |
| 2151 | Dead, transmitter, awful | 2146 |
| 8528 | Hard, top, lap, scroll, pad | 2151 |
| 10A63 | Outstanding, playback, release, model | 10A70 |
| 10A70 | Gamer, thrive bigger, hub | 10A63 |
| 10B3E | Toshiba, lifetime, replacement, warranty | 10A70 |
| 21091 | Foot, pace, distance, run | 10B3E |
| 79A0D | Heavy, metal, cover, bottom | 21091 |
| 10C0C3 | Traffic, highway, city, road | 79A0D |
| 214C93 | Bluetooth, reception, works, feedback | 214C90 |
| 214C90 | Bright, display, control | 214C93 |

each other through structural associations. Structures are preserved through iteratively split-
ting the communities into sub-communities. The communities that are separated in later iter-
ations are considered closer as compared to the communities that are separated in the former
iterations. The structural position of each community in the graph of all prior data is derived
through subsequent splits, and is presented as hexadecimal codes as IDs in Table 5. The com-
munities with a smaller difference of hexadecimal codes indicate a nearer common parent and
a closer relationship and vice versa. These values can be integers or binary but we have pre-
sented them as hexadecimal values for compactness. Fig 2 shows the first five iterations of
recursively splitting communities into sub-communities. The communities with grey back-
ground will continue with the process. The others with white background have reached their
stopping criteria of either too few nodes or too expensive cut to continue with.

In the third step, the communities extracted are evaluated for their quality. Communities
with weakly associated nodes cannot effectively communicate their impact and are therefore
filtered out. Only 84 communities were left behind after the *WGCI* and entropy based filtering.
The threshold values for *WGCI* and entropy were set to 1 and 0.2, respectively. These values
are experimentally evaluated for the given dataset. Dropping the thresholds add to computa-
tional cost without showing reasonable improvement in results. Table 5 shows the structural
information and the top few words for 20 sample rules. Table 6 shows other properties, i.e.,
size, entropy, Lins approximation, *WGCI* and average degree node across community for the
corresponding rules in Table 5. These evaluation scores are used to retain only good quality
communities with higher compactness within its nodes while filter others.

In the fourth step, the communities are transformed and represented as rules. The hub
node in the community is represented as rule head. The Lins Approximation value of each
community is represented as the impact of its corresponding rule. It refers to the bias with
which the inference technique is manipulated in favor of the rule by using generalized polya

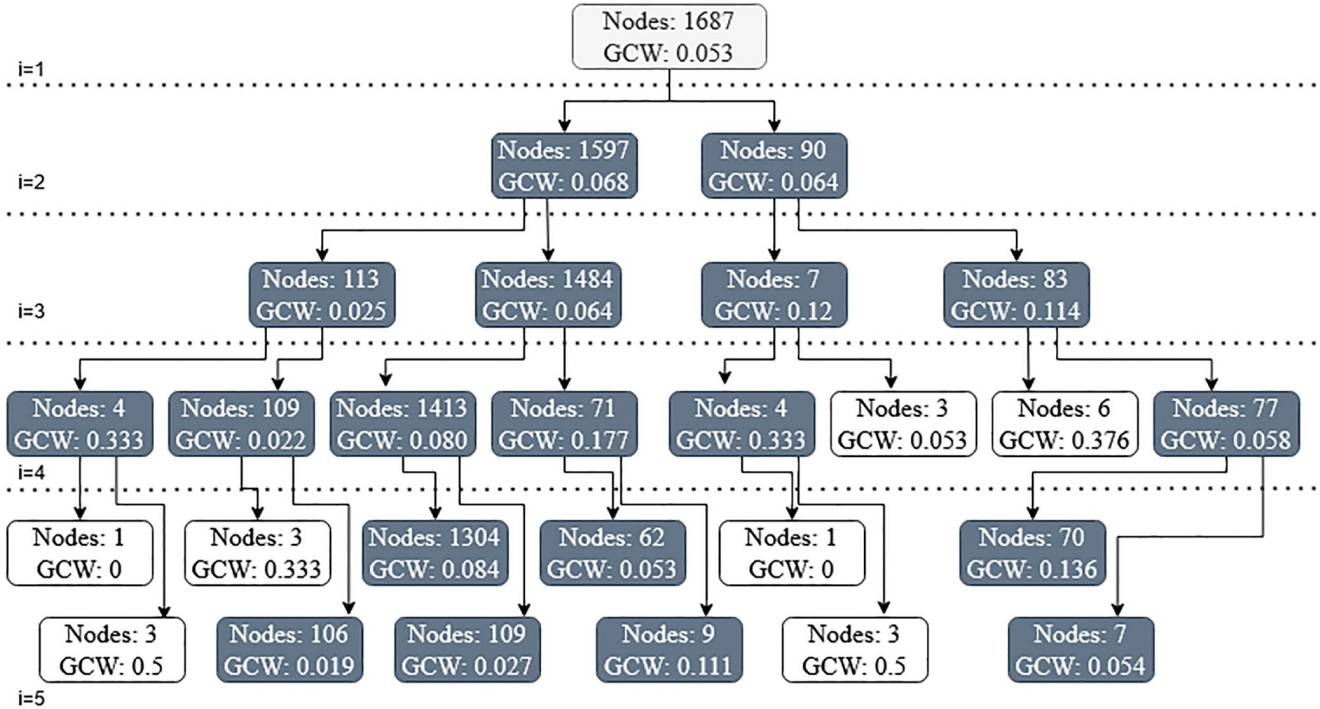

**Fig 2. Hierarchical breakdown of graph into communities till first 5 levels, nodes per community and their respective graph cut weight.**

**Table 6. Evaluation scores of the communities shown in Table 5.**

| Comm Sr.No. | Size | Entropy | Lins Approx. | WGCI | Avg. Node Degree |
|---|---|---|---|---|---|
| 1 | 7 | 2.787 | 1.8 | 7.073 | 56.893 |
| 2 | 13 | 3.816 | 3.249 | 0.236 | 12.153 |
| 3 | 5 | 1.108 | 1.249 | 0.020 | 2 |
| 4 | 5 | 1.130 | 1.249 | 0.040 | 4.6 |
| 5 | 6 | 1.440 | 1.499 | 0.040 | 3.83 |
| 6 | 5 | 1.055 | 1.249 | 0.030 | 2.4 |
| 7 | 8 | 2.087 | 1.999 | 0.061 | 4.125 |
| 8 | 5 | 1.085 | 1.249 | 0.054 | 6.6 |
| 9 | 6 | 1.425 | 1.499 | 0.86 | 7.83 |
| 10 | 5 | 1.135 | 1.249 | 0.336 | 19.2 |
| 11 | 6 | 1.449 | 1.499 | 0.036 | 3.33 |
| 12 | 28 | 1.745 | 1.749 | 0.0833 | 6.85 |
| 13 | 6 | 1.488 | 1.499 | 0.0923 | 10.83 |
| 14 | 5 | 1.109 | 1.249 | 0.0414 | 3.8 |
| 15 | 5 | 1.071 | 1.249 | 0.3227 | 48.6 |
| 16 | 7 | 1.698 | 1.749 | 1.538 | 103.285 |
| 17 | 5 | 1.129 | 1.249 | 0.348 | 52.8 |
| 18 | 7 | 1.817 | 1.749 | 0.829 | 52.85 |
| 19 | 5 | 1.141 | 1.249 | 0.062 | 6 |
| 20 | 4 | 1.746 | 1.448 | 0.085 | 75.4 |

urn model. Step five is related to transferring the impact of rules into the new task. It can be observed that most of the rules have their sizes ranging from 4 to 7 nodes as the natural arrangement of words in the given data. It may vary for other datasets and therefore, the learning mechanism is kept flexible to adapt to it.

The quality of the topics extracted is evaluated using topic coherence. It presents the association or correlation among the words of a topic [71]. Thus, the words of a topic that have high contextual correlation results in a higher topic coherence score for the topic. It is the dominant topic evaluation technique that has the highest relevance to human judgment. The topic coherence for a topic $k$ can be calculated as,

$$TC_k = \sum_{w,w' \in T_k} log(P(w|w')) \tag{10}$$

where $T_k$ represents the $k^{th}$ topic in the given corpus while $w$ and $w'$ are two distinct words belonging to the topic $T_k$. It is the double sum of log of conditional probabilities across all the words in the topic. The top 30 words with highest probability in a topic are used to calculate topic coherence as in literature [71]. The topic coherence of a dataset is determined by averaging the topic coherence over all the topics in a given dataset.

The learning curve of lifelong topic modeling is evaluated based on improvement in the quality of topics as topic coherence. Fig 3 shows the comparison of NHLTM with some of the existing approaches including lifelong topic modeling (LTM) [31], automatic must link cannot link (AMC) [34], automatic must link cannot link with must links only (AMC-M) [34], online learning based automatic must link cannot link (OAMC) [12] and the basic latent dirichlet allocation approach. All approaches are provided with the same experience in the form of 10% of the data to mine rules from. They are then presented with a new category as current data to extract interpretable topics by minimizing the impact of noise in the new categories with the help of the mined rules. The top 30 words in a topic are used to calculate the topic coherence of a topic. The NHLTM approach has resulted in consistently improving the topic coherence across all the datasets in comparison to others.

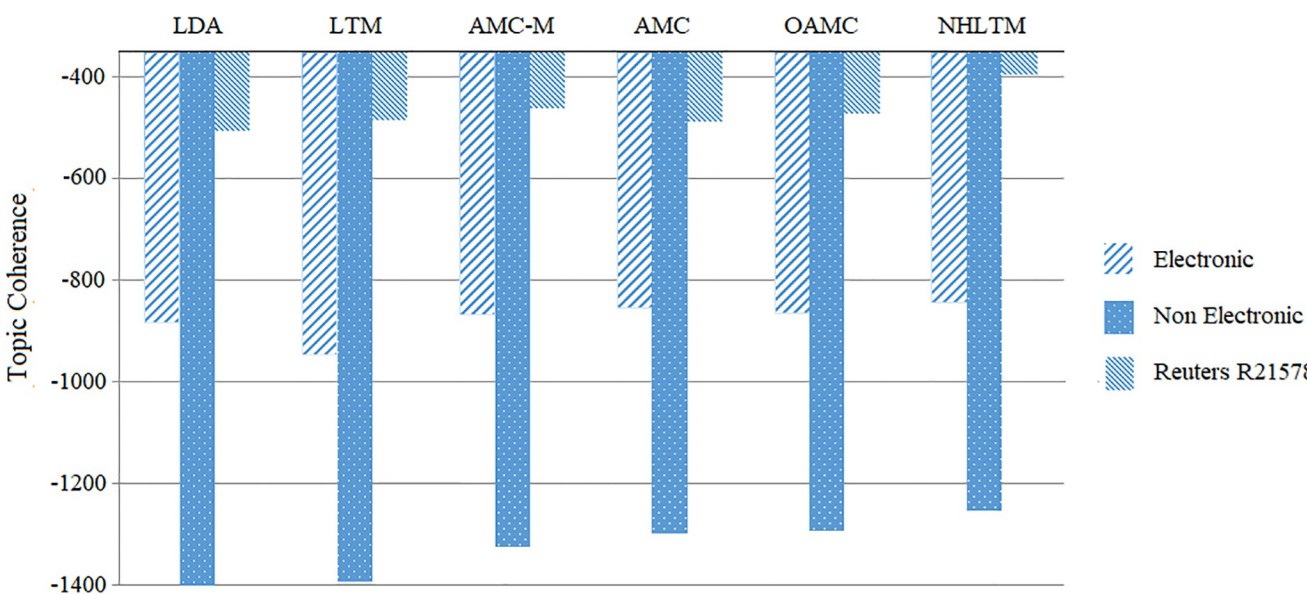

**Fig 3. Comparison of NHLTM with existing approaches.**

A hierarchy of topics is extracted for each dataset that shows the concepts discussed within the dataset at different granularity levels. The hierarchies are compared in terms of their topic coherence at different levels in the hierarchy as the number of topics increases. The empirical log-likelihood indicates generality of the models as its ability to estimate the unseen data. Topic coherence of the proposed NHLTM approach shows higher improvement down the hierarchy from level one to level six, in comparison to the HLDA approach. The topics at higher levels represent generic concepts and therefore attain lower topic coherence. The lower levels have specific concepts with compact representation. The given results indicate the compactness of topics produced by NHLTM at all levels of the hierarchy. The value of empirical log-likelihood should preferably increase or stabilize as the model makes better predictions of the unseen data. The empirical log-likelihood values produced by NHLTM model are maintained lower than HLDA and are also depicted in Fig 4.

The topics of the *Alarm Clock* dataset are more scattered having more branches at the lower level in Fig 5. A possible reason is that the users discussing topics of *Alarm Clock* referred to the same topics using a variety of expressions. Discussing more topics within a document leads a closer relation among those topics. The topics *Noise wake* and *Design flaw* have very weak association with all the other topics and are therefore, directly linked to the root at the highest level. The root represents the domain of the dataset, which is *Alarm Clock* in this case. The topics *Small bedside* and *Easy cheap* are closely related. The other topics that are closely related, represent the operational topics as *Light feature, Simple stuff* and *Digital number*. The topics *Radio instructions* and *Display control* represent the soft features or controls of the product. These topics are linked to the previous group of topics as operations and controls are somewhat closely related. The topics *Battery star* and *Plastic piece* doesn't have a very clear sense but loosely represent the exterior of the product. The next few topics also have weaker association. The non-electronic products like *Cars, Boats* and *Food* would have more diverse vocabulary as compared to electronic products and therefore, their topic hierarchies could be more easily comprehendible.

## Conclusion

Lifelong topic models enable the conventional topic model to learn topics continuously from the knowledge accumulated from the past which is updated regularly based on new information. On the other hand, hierarchical topic modeling extends topic modeling for grouping into a hierarchical structure. This study combines the two and proposes hierarchical lifelong topic models which allows the examination of topics at different levels of granularity and continuously adjust the granularity of the topics as more information is made available. The existing hierarchical topic models lack the lifelong learning mechanism and therefore, cannot generate topic hierarchies from the word associations only. On the other hand, the existing LTM approaches do not attempt to generate topic hierarchies. NHLTM has introduced spectral clustering and preserve the communities as rules along with their associations that contribute towards extracting topic hierarchies. A key issue in hierarchical lifelong topic modeling is to extract rules that will not only preserve the hierarchical information among the rules but will also regularly update them as information evolves. We introduced a network community based hierarchical lifelong topic models or NHLTM to address this issue. Experimental results on the datasets of Chen 2014 electronic, Chen 2014 non-electronic and R21578 Reuters indicate improvement of the hierarchical topic structures for maintaining high topic coherence that increases from general topics to specific topics.

The proposed approach has opened up doors for further research. More specifically, one may extend the NHLTM approach to include topics with multiple parents and to consider

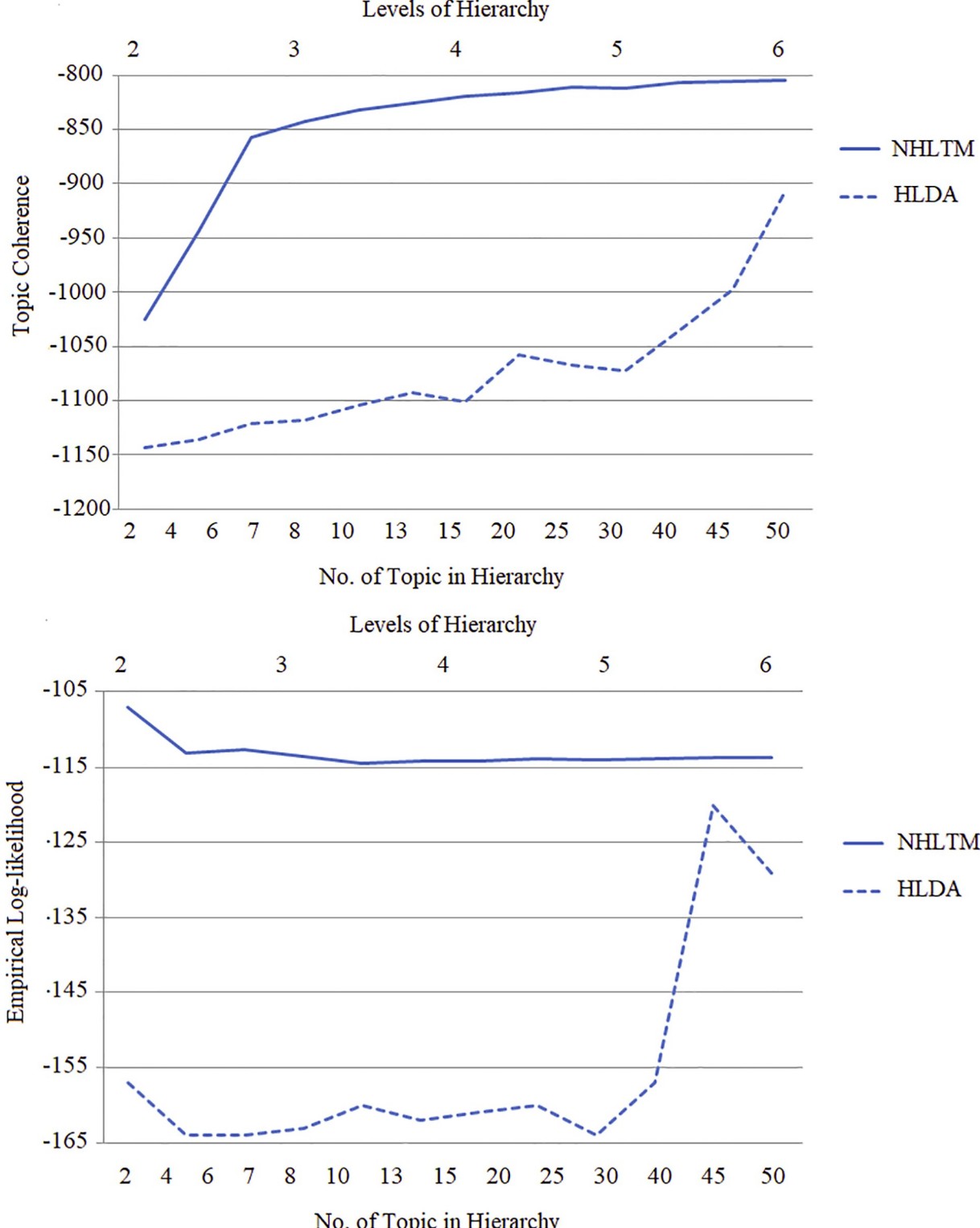

**Fig 4. Comparison of NHLTM and HLDA with increasing number of topics and levels of hierarchy for the Alarm clock dataset in large-scale data.**

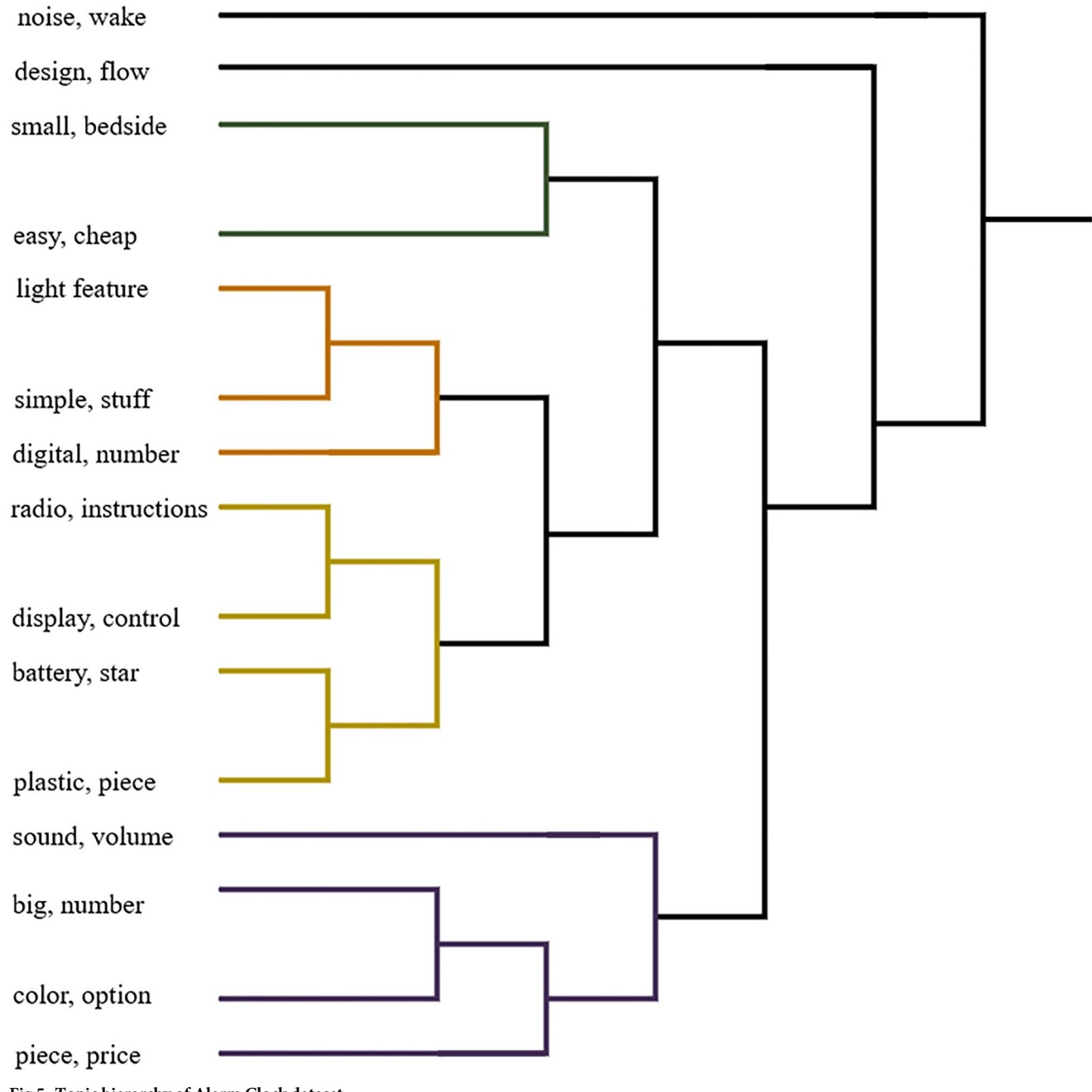

**Fig 5. Topic hierarchy of Alarm Clock dataset.**

polysemy at the rules level. Moreover, the effect of using embeddings in the rule mining component and its impact on the quality of extracted hierarchy may provide further useful insights.

## Author Contributions

**Conceptualization:** Muhammad Taimoor Khan, Nouman Azam.

**Formal analysis:** Muhammad Taimoor Khan, Furqan Aziz.

**Investigation:** Muhammad Taimoor Khan.

**Methodology:** Muhammad Taimoor Khan, Furqan Aziz.

**Project administration:** Shehzad Khalid.

**Supervision:** Shehzad Khalid.

**Writing – original draft:** Muhammad Taimoor Khan, Nouman Azam.

**Writing – review & editing:** Nouman Azam.

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
