## [Decision Letter · Decision Letter 0]

12 Nov 2020

PONE-D-20-29449

Hierarchical Lifelong Topic Modeling Using Rules Extracted From Network Communities

PLOS ONE

Dear Dr. Khan,

Thank you for submitting your manuscript to PLOS ONE. After careful consideration, we feel that it has merit but does not fully meet PLOS ONE’s publication criteria as it currently stands. Therefore, we invite you to submit a revised version of the manuscript that addresses the points raised during the review process.

We look forward to receiving your revised manuscript.

Kind regards,

Diego Raphael Amancio

Academic Editor

PLOS ONE

Journal Requirements:

"This work is partially supported by faculty research support fund of National

University of Computer and Emerging Sciences (NUCES), Pakistan."

2.1. We note that you have provided funding information that is not currently declared in your Funding Statement. However, funding information should not appear in the Acknowledgments section or other areas of your manuscript. We will only publish funding information present in the Funding Statement section of the online submission form.

 [xxx]

2.2. Please provide an amended statement that declares *all* the funding or sources of support (whether external or internal to your organization) received during this study, as detailed online in our guide for authors at http://journals.plos.org/plosone/s/submit-now.  Please also include the statement “There was no additional external funding received for this study.” in your updated Funding Statement.

Reviewers' comments:

Reviewer's Responses to Questions

**Comments to the Author**

1. Is the manuscript technically sound, and do the data support the conclusions?

Reviewer #1: Partly

Reviewer #2: Partly

2. Has the statistical analysis been performed appropriately and rigorously? 

Reviewer #1: No

Reviewer #2: Yes

3. Have the authors made all data underlying the findings in their manuscript fully available?

Reviewer #1: Yes

Reviewer #2: No

4. Is the manuscript presented in an intelligible fashion and written in standard English?

Reviewer #1: No

Reviewer #2: Yes

5. Review Comments to the Author

Reviewer #1: This paper presents a hierarchical lifelong topic model called NHLTM. This new model combines lifelong learning, with network community detection and topic modelling.

Overall I found that the manuscript was well organized and the contributions were clear. There were several non-native grammar errors. I probably did not catch them all, but I did upload my notes to this review. Please look at my ink-notes for detailed comments.

My broad remarks are as follows:

1. Reporting raw likelihood values is not normal and not meaningful. Please do not report -856, etc. as results. Just say that it was improved.

2. Section 1 and section 2 are otherwise well organized with only minor typographical and grammatical errors.

3. In section 4 you talk about treating the text corpora as a graph. There exists the hierarchical document topic model (HDTM) which does something similar here that you could look at.

4. The input of Alg 1, does it require preprocessing of the whole data each time? The input looks like it does.

5. In general, Section 4 is too verbose. You do not need to describe each line of Alg 1. Alg 2, Alg 3, and Alg 4 are very straightforward and can just be described in prose. Likewise Eq 5 and Eq 6 are well known functions. Just say normalized Laplacian and Fiedler Vector. You don't need to derive it. Any reader will know what those things are. Eq 7 is called "conductance cut" and is well known too - there is no need to describe these things. I think that Section 4 could be reduced to 3-4 pages total.

6. In section 5, you only perform experiments on a single dataset. This is not enough for conclusive study. There exist enormous datasets for study. You need to perform a much more thorough analysis with many datasets.

7. In Table 5, what is the hexadecimal data supposed to show? Why not just represent things as ints? What is Comm Sr. No. column?

8. Fig 2 us supposed to have some nodes with a grey background, but my printout does not have anything in grey.

9. In Section 5, where do you test the "lifelong" portion of NHLTM?

10. Whats the purpose of Table 6? And on page 23 generally, raw values are just not informative.

11. Fig 3 has the y-axis flipped. It goes from largest on the bottom to smallest on the top. This is quite abnormal.

12. Table 7 should be a figure.

13. Conclusions are not strong. What did we learn in this paper? What are the limitations of NHLTM? What is the future work. Expand this section please.

Overall I think the technical contribution of this paper is ok, but the experiments are severely lacking. Significant additional work is needed before this paper can be accepted.

Reviewer #2: While interesting, I believe this paper needs major revisions for improved clarity and a more succinct narrative.

- What do you mean by rules? You should present a clear example and definition very early. In one case, it appears that a rule consists of a correlation of two words, but a subsequent equation seems to contradict this.

- Am I right in thinking you used raw frequency? In "classic" approaches to topic modeling, raw frequency is rarely used. It's more common to see something like positive pointwise mutual information (PPMI). Were alternatives to raw frequency explored? If so, why wasn't this reported?

- Moreover, typically, term term matrices have a zero diagonal.

- I recommend that you avoid using "Extract communities using Algorithm 3" when you can instead reference your favored technique by name.

- Your tables and figures all warrant real captions so that they can be digested on their own.

- Word embeddings have revolutionized natural language processing. Recent approaches to topic modeling that have appeared in venues such as TACL and various ACL conferences have all involved word embeddings to some degree. Why is no discussion or comparison made here? While this might require a second evaluation on another dataset, I think it's necessary.

- Speaking of recent approaches, how do neural topic models (ex. VAE-based) compare to your approach? For example, a comparison to Gupta et al's Neural Topic Modeling with Continual Lifelong Learning (ICML 2020) would be nice.

- How does the approach work with different community detection algorithms? Why did you select this particular one without any comparison?

- How are you evaluating the lifelong aspect of your algorithm?

- AMC-M, AMC and OAMC

- Are the abbreviated forms of these acronyms introduced anywhere?

- Table 5: this should be reformatted to make it more obvious which communities are "close".

- Please make your code available for reproducibility and future comparisons. Doing so is now standard practice. Ideally, I would like to see how you compared the different models.

- Though not a serious problem, there are numerous typographical errors (repeated words, inconsistent formatting, agreement errors, missing determiners, etc.).

6. PLOS authors have the option to publish the peer review history of their article (what does this mean?). If published, this will include your full peer review and any attached files.

Reviewer #1: No

Reviewer #2: No

---

## [Author Response · Author response to Decision Letter 0]

1 Jan 2021

We are thankful to the reviewers for taking time to review the paper and liking it. We tried to incorporate all the concerns of the reviewers and we hope that the revisions in the paper will meet the expectations of the reviewer. Please find detailed responses and the resulting changes in the manuscript in the ResponseNHLTMv4 document attached.

---

## [Decision Letter · Decision Letter 1]

8 Jun 2021

PONE-D-20-29449R1

Hierarchical Lifelong Topic Modeling Using Rules Extracted From Network Communities

PLOS ONE

Dear Dr. Khan,

Thank you for submitting your manuscript to PLOS ONE. After careful consideration, we feel that it has merit but does not fully meet PLOS ONE’s publication criteria as it currently stands. Therefore, we invite you to submit a revised version of the manuscript that addresses the points raised during the review process.

The reviewer pointed out that major issues have been addressed. However, language should be checked in order to improve readability. 

We look forward to receiving your revised manuscript.

Kind regards,

Diego Raphael Amancio

Academic Editor

PLOS ONE

Journal Requirements:

Reviewers' comments:

Reviewer's Responses to Questions

**Comments to the Author**

1. If the authors have adequately addressed your comments raised in a previous round of review and you feel that this manuscript is now acceptable for publication, you may indicate that here to bypass the “Comments to the Author” section, enter your conflict of interest statement in the “Confidential to Editor” section, and submit your "Accept" recommendation.

Reviewer #1: All comments have been addressed

2. Is the manuscript technically sound, and do the data support the conclusions?

Reviewer #1: Yes

3. Has the statistical analysis been performed appropriately and rigorously? 

Reviewer #1: Yes

4. Have the authors made all data underlying the findings in their manuscript fully available?

Reviewer #1: Yes

5. Is the manuscript presented in an intelligible fashion and written in standard English?

Reviewer #1: Yes

6. Review Comments to the Author

Reviewer #1: The author has addressed all of my comments. The new parts of the paper need a thorough proofread - there are minor grammatical issues, but nothing major.

7. PLOS authors have the option to publish the peer review history of their article (what does this mean?). If published, this will include your full peer review and any attached files.

Reviewer #1: No

---

## [Author Response · Author response to Decision Letter 1]

1 Jul 2021

Comments by Reviewer 1

The author has addressed all of my comments. The new parts of the paper need a thorough proofread - there are minor grammatical issues, but nothing major.

Response: 

We are thankful to the reviewer for accepting the modifications we have made to the manuscript in response to the comments provided. We have fixed the grammatical issues as highlighted by the reviewer and have proof read the document multiple times. 

Below are some of the updates that we made with regards to the grammar in this version of the manuscript.

Some examples of correct usage of the article. 

‘….50 electronic product categories, Chen 2014 dataset …’ changed to 

‘….50 electronic product categories, the Chen 2014 dataset …’

‘Filtering out edges with smaller weights and nodes …’ changed to

‘The filtering out edges with smaller weights and nodes …’

‘A higher value indicates better correlation among the words of a topic.’ changed to

‘A higher value indicates a better correlation among the words of a topic. ’

‘All approaches are provided same experience in the … ’ changed to

‘All approaches are provided the same experience in the …’

‘Top 30 words in a topic are used to calculate the topic coherence of a topic.’ changed to

‘The top 30 words in a topic are used to calculate the topic coherence of a topic.’

‘The communities with smaller difference … ‘ changed to

‘The communities with a smaller difference …’

‘ … in comparison to HLDA approach.’ Changed to

‘ … in comparison to the HLDA approach.

‘ … The given results indicate compactness …’ changed to

‘ … The given results indicate the compactness …’

Some Examples of adding comma at the right place for better readability

‘In this paper we examine a new type of topic model …’ changed to

‘In this paper, we examine a new type of topic model …’

‘… meaningful words as verbs, adverbs, adjectives and nouns.’ changed to

‘… meaningful words as verbs, adverbs, adjectives, and nouns.’

‘… case of the first two datasets i.e., Chen 2014 Electronic …’ changed to

‘…case of the first two datasets, i.e., Chen 2014 Electronic …’

‘… having documents in thousands, hundreds and below that.’ changed to

‘… having documents in thousands, hundreds, and below that.’

‘…is derived through subsequent splits and …’ changed to

‘…is derived through subsequent splits, and …’ 

‘Table 6 shows other properties i.e., size, entropy …’ changed to

‘Table 6 shows other properties, i.e., size, entropy …’

‘ … generic concepts and therefore, attain …’ changed to

‘ … generic concepts and therefore attain …’

Some Examples of preposition correction

‘… has the highest relevance with human judgment.’ changed to

‘… has the highest relevance to human judgment.’

‘… impact of noise in the new categories with the help the mined rules.’ changed to

‘…impact of noise in the new categories with the help of the mined rules’

‘ … All approaches are provided same experience in the …’ changed to

‘ … All approaches are provided with same experience in the …’

Some examples of correct singular/plural usage 

‘… are directly linked to the root of the hierarchy as an outlier’ changed to

‘…topics link to the root of the hierarchy as outliers’

‘…extracting topic hierarchies and is processed …’ changed to

‘…extracting topic hierarchies and are processed …’

‘These type of classes are frequently encountered in …’ changed to

‘These types of classes are frequently encountered in …’

Some examples of correct form of verb 

‘Lifelong topic models enables the conventional …’ changed to

‘Lifelong topic models enable the conventional …’

‘This study combines the two and proposed a hierarchical …’ changed to

‘This study combines the two and proposes a hierarchical …

‘… word embeddings has improved various natural … ’ changed to

‘…word embeddings have improved various natural …’

‘… topic models on word embeddings help better grouping… ’ changed to

‘… topic models on word embeddings helps better grouping…’

‘… the communities based on new information which result in updated … ’ changed to

‘…the communities based on new information which results in updated …’

‘… computing eigenvalues that allows considering …’ changed to

‘…computing eigenvalues that allow considering …’

‘ … of information and has dimensionally reduction …’ changed to

‘ … of information and have dimensionally reduction …’

Some examples of improving the structure of sentence for easy to follow understanding. 

‘Moreover, such support is required each time when new information is processed in the form of new dataset.’ changed to

‘Moreover, each time new information is available as a dataset, the same support is required.’

‘Hierarchies are highly important in displaying the various components of a system in a tree like structure that has generic concepts at higher levels while specific concepts at lower levels [33].’ changed to

‘Hierarchies are crucial in displaying the various components of a system in a tree-like structure. It has generic concepts at higher levels, while specific concepts are at lower levels [33].’

‘Filtering out edges with smaller weights and nodes with fewer edges, the graph is refined for efficient analysis. Thus we have a smaller but more densely connected network after the filtering phase.’ changed to

‘The filtering mechanism refines the graph for efficient analysis. It drops the nodes and edges that have a low degree and edge weight, respectively. Thus we have a smaller but more densely connected network after the filtering phase.’

‘In this section, we present detailed experimental results of the proposed NHLTM model for extracting topic hierarchies for a sequence of tasks.’ changed to

‘In this section, we present detailed experimental results of the proposed NHLTM model for a sequence of tasks.’

‘Aggressive filtering is applied to remove weak nodes and their edges as they cannot be part of compact communities which also results in the average edge weight.’ changed to

‘Through aggressive filtering, the weak nodes and edges are removed as they are least likely to be part of compact communities. It results in increasing the average node degree and edge weight.’

At the end, we would like to pay thanks again to the editor in chief, associate editor and the reviewers. Indeed, their constructive comments helped to improve the overall quality of the manuscript.

---

## [Decision Letter · Decision Letter 2]

30 Dec 2021

PONE-D-20-29449R2Hierarchical Lifelong Topic Modeling Using Rules Extracted From Network CommunitiesPLOS ONE

Dear Dr. Khan,

Thank you for submitting your manuscript to PLOS ONE. After careful consideration, we feel that it has merit but does not fully meet PLOS ONE’s publication criteria as it currently stands. Therefore, we invite you to submit a revised version of the manuscript that addresses the points raised during the review process.

We look forward to receiving your revised manuscript.

Kind regards,

Diego Raphael Amancio

Academic Editor

PLOS ONE

Journal Requirements:

Reviewers' comments:

Reviewer's Responses to Questions

**Comments to the Author**

1. If the authors have adequately addressed your comments raised in a previous round of review and you feel that this manuscript is now acceptable for publication, you may indicate that here to bypass the “Comments to the Author” section, enter your conflict of interest statement in the “Confidential to Editor” section, and submit your "Accept" recommendation.

Reviewer #1: All comments have been addressed

Reviewer #3: (No Response)

2. Is the manuscript technically sound, and do the data support the conclusions?

Reviewer #1: Yes

Reviewer #3: Yes

3. Has the statistical analysis been performed appropriately and rigorously? 

Reviewer #1: Yes

Reviewer #3: Yes

4. Have the authors made all data underlying the findings in their manuscript fully available?

Reviewer #1: Yes

Reviewer #3: (No Response)

5. Is the manuscript presented in an intelligible fashion and written in standard English?

Reviewer #1: Yes

Reviewer #3: Yes

6. Review Comments to the Author

Reviewer #1: Grammatical changes appear to be satisfactory. I have no further comments to make regarding the revision of this paper.

Reviewer #3: This paper proposes an interesting approach for topic modeling. More specifically, the authors put together concepts of NLP and network science to develop a topic modeling approach. Since the author already answered the other reviewers, I have only some points concerning the text.

- Since the method involves techniques from different areas, I believe it would be better to explain in more detail some basic concepts. For example, how to calculate and interpret "topic coherence" and "Von Neumann entropy". Note that these concepts are simple, but researchers only from one of the areas involved in this work would not know these concepts.

- On page 12, the authors describe that a co-occurrence graph is used. However, it would help the reader to describe more characteristics of this graph. More specifically, if the edges are directed and/or weighted and how it is determined.

- Topic modeling based on communities of complex networks have been studied in:

Silva FN, Amancio DR, Bardosova M, Costa LD, Oliveira Jr ON. Using network science and text analytics to produce surveys in a scientific topic. Journal of Informetrics. 2016 May 1;10(2):487-502.

Furthermore, a multi-scale extension of this method was proposed in:

Ceribeli C, de Arruda HF, da Fontoura Costa L. How coupled are capillary electrophoresis and mass spectrometry?. Scientometrics. 2021 May;126(5):3841-51.

Considering that both papers are significantly related to the paper, I think these references could be considered.

- In step 5 of section 4.2, it is written: "The α as the smoothing factor." I think that it should be: "α is the smoothing factor." Furthermore, in "Moreover, the β is used as a smoothing factor", I think there is no "the".

7. PLOS authors have the option to publish the peer review history of their article (what does this mean?). If published, this will include your full peer review and any attached files.

Reviewer #1: No

Reviewer #3: No

---

## [Author Response · Author response to Decision Letter 2]

15 Jan 2022

Comments by Reviewer 1

Grammatical changes appear to be satisfactory. I have no further comments to make regarding the revision of this paper.

Response: 

We are thankful to the reviewer for his valuable time and feedback. The suggestions were very helpful and contributed in improving the overall quality of our manuscript.

Comments by Reviewer 3

This paper proposes an interesting approach for topic modeling. More specifically, the authors put together concepts of NLP and network science to develop a topic modeling approach. Since the author already answered the other reviewers, I have only some points concerning the text.

Response:

We are thankful to the reviewer for taking time to review the paper and liking it. We tried to incorporate all the concerns of the reviewer and we hope that the revisions in the paper will meet the expectations of the reviewer. A detailed comment by comment response is provided in a separate document named "response to reviewers". 

We are very thankful to the reviewers and editor for contributing to improve the quality of this manuscript.

---

## [Decision Letter · Decision Letter 3]

14 Feb 2022

Hierarchical Lifelong Topic Modeling Using Rules Extracted From Network Communities

PONE-D-20-29449R3

Dear Dr. Khan,

We’re pleased to inform you that your manuscript has been judged scientifically suitable for publication and will be formally accepted for publication once it meets all outstanding technical requirements.

Kind regards,

Jerry Chun-Wei Lin

Academic Editor

PLOS ONE

Additional Editor Comments (optional):

Reviewers' comments:

Reviewer's Responses to Questions

**Comments to the Author**

1. If the authors have adequately addressed your comments raised in a previous round of review and you feel that this manuscript is now acceptable for publication, you may indicate that here to bypass the “Comments to the Author” section, enter your conflict of interest statement in the “Confidential to Editor” section, and submit your "Accept" recommendation.

Reviewer #1: All comments have been addressed

2. Is the manuscript technically sound, and do the data support the conclusions?

Reviewer #1: Yes

3. Has the statistical analysis been performed appropriately and rigorously? 

Reviewer #1: Yes

4. Have the authors made all data underlying the findings in their manuscript fully available?

Reviewer #1: Yes

5. Is the manuscript presented in an intelligible fashion and written in standard English?

Reviewer #1: Yes

6. Review Comments to the Author

Reviewer #1: Thank you for addressing my reviews. The current version of the paper does not pose any obvious flaws according to the PLOS ONE publication model.

7. PLOS authors have the option to publish the peer review history of their article (what does this mean?). If published, this will include your full peer review and any attached files.

Reviewer #1: No

---

## [Editor Report · Acceptance letter]

23 Feb 2022

PONE-D-20-29449R3 

Hierarchical Lifelong Topic Modeling Using Rules Extracted From Network Communities 

Dear Dr. Khan:

I'm pleased to inform you that your manuscript has been deemed suitable for publication in PLOS ONE. Congratulations! Your manuscript is now with our production department. 

Kind regards, 

on behalf of

Prof. Jerry Chun-Wei Lin 

Academic Editor

PLOS ONE